# Superposing Many Tickets into One:
# A Performance Booster for Sparse Neural Network Training

**Lu Yin**[1]    **Vlado Menkovski**[1]    **Meng Fang**[1]    **Tianjin Huang**[1]    **Yulong Pei**[1]    **Mykola Pechenizkiy**[1]

**Decebal Constantin Mocanu**[2,1]                    **Shiwei Liu**[1]

[1]Eindhoven University of Technology , Eindhoven, the Netherlands
[2]University of Twente, Enschede, the Netherlands

## Abstract

Recent works on sparse neural network training have shown that a compelling trade-off between performance and efficiency can be achieved. Existing sparse training methods usually strive to find the best sparse subnetwork possible in one single run, without involving any expensive dense or pre-training steps. For instance, dynamic sparse training (DST), as one of the most prominent directions, is capable of reaching a competitive performance of dense training by iteratively evolving the sparse topology during the course of training. In this paper, we argue that it is better to allocate the limited resources to create multiple low-loss sparse subnetworks and superpose them into a stronger one, instead of allocating all resources entirely to find an individual subnetwork. To achieve this, two desiderata are required: (1) efficiently producing many low-loss subnetworks, the so-called cheap tickets, within one training process limited to the standard training time used in dense training; (2) effectively superposing these cheap tickets into one stronger subnetwork without going over the constrained parameter budget. To corroborate our conjecture, we present a novel sparse training approach, termed **Sup-tickets**, which can satisfy the above two desiderata concurrently in a single sparse-to-sparse training process. Across various models on CIFAR-10/100 and ImageNet, we show that Sup-tickets integrates seamlessly with the existing sparse training methods and demonstrates consistent performance improvement.

## 1  INTRODUCTION

Over the past years, large-scale deep learning models with billions, even trillions of parameters have improved the

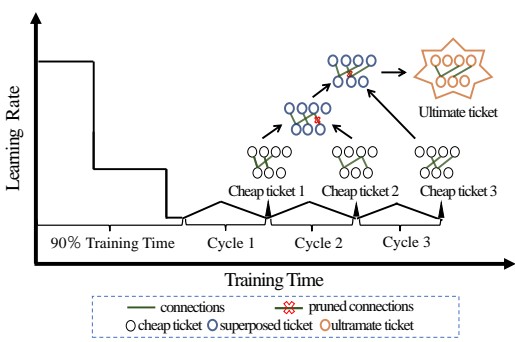

Figure 1: The schematic view of Sup-tickets. Multiple subnetworks (cheap tickets) are efficiently produced within the last 10% of the training time and are superposed into one single subnetwork with boosting performance while maintaining the target sparsity. We term the "ultimate ticket" as the final subnetwork used for inference.

state-of-the-art in nearly every downstream task [Shoeybi et al., 2019, Radford et al., 2021, Fedus et al., 2021]. The compelling results achieved by these large-scale models motivate researchers to pursue increasingly gigantic models without thinking too much about the limited resources of our planet. Fortunately, many prior techniques for neural network acceleration have already been proposed, which can effectively trim down the memory requirements and computational costs while retaining high accuracy [Mozer and Smolensky, 1989, Han et al., 2015, Gale et al., 2019].

Among them, sparse neural network training [Mocanu et al., 2018, Evci et al., 2020, Bellec et al., 2018] stands out and receives growing attention recently due to its high efficiency in both the training and inference phases. Instead of inheriting well-performing sparse networks from a trained dense network, sparse training approaches typically start from a randomly initialized sparse network and only require training a subset of the corresponding dense network. Since

*Accepted for the 38th Conference on Uncertainty in Artificial Intelligence* (UAI 2022).

this sparse-to-sparse training process does not involve any dense or pre-training steps, the memory requirements and the floating-point operations (FLOPs) are only a fraction of the traditional dense training. Nonetheless, naively training a sparse neural network from scratch leads to poor solutions in general compared with training a dense network [Evci et al., 2019]. Dynamic sparse training (DST) [Mocanu et al., 2018] significantly improves the trainability of sparse networks by dynamically exploring new connectivities during training, while maintaining the fixed parameter count. Compared with methods that train with the fixed sparse connectivity [Mocanu et al., 2016, Lee et al., 2018], DST substantially improves the expressibility of sparse networks, and thus leads to better generalization performance [Liu et al., 2021c]. However, the accuracy of extremely sparse subnetworks (e.g., at sparsity[1] 95% or 90%) usually remains below the full dense training under a regular training epoch number [Evci et al., 2020, Liu et al., 2021b]. Enabling sparse training at extreme sparsities to match or even surpass the performance of dense training under a typical amount of training epochs will significantly benefit sparse training in practice.

Increasingly more evidence on sparse training [Liu et al., 2021a] and dense training [Garipov et al., 2018, Draxler et al., 2018, Fort and Jastrzebski, 2019] reveal that many independent local optima exist in different low-loss basins of the loss landscape. Inspired by these observations, we go one step further to pursue an approach that can boost the performance of sparse training by leveraging these widely-existing low-loss basins. Specifically, we propose Superposing Tickets, or briefly **Sup-tickets**, which could produce many subnetworks (cheap tickets) in one single run and then superposes all of them into one at the same sparsity. Doing so allows us to leverage the knowledge from various well-performing cheap tickets, while still maintaining the training and inference efficiency of sparse training. Overall, we summarize our contributions below:

- We propose Sup-tickets, a novel sparse training approach that produces and superposes many cheap yet well-performing subnetworks (cheap tickets) during one sparse-to-sparse training run. The ultimate superposed subnetwork achieves stronger results in predictive accuracy and uncertainty estimation while maintaining the target sparsity.

- Sup-tickets is a general and versatile performance booster for sparse training, which seamlessly integrates with other state-of-the-art sparse training methods. We conduct extensive experiments to evaluate our method. Across various popular architectures on CIFAR-10/100 and ImageNet, Sup-tickets improves the performance of various sparse training methods without extending

---

[1]The term sparsity refers to the proportion of the neural network's weights that are zero-valued.

the training time.

- More impressively, in conjunction with the advanced sparse training methods – GraNet [Liu et al., 2021b], Sup-tickets boosts the performance of sparse training over the dense training on CIFAR-10/100 at extreme sparsity levels around 90% ∼ 95%, enhancing the great potentials of sparse training in practice.

## 2 RELATED WORK

### 2.1 SPARSE NEURAL NETWORK TRAINING

Sparse neural network training is a thriving topic. It aims to train initial sparse neural networks from scratch and chase competitive performance with their dense counterparts, while using only a fraction of resources of the latter. According to whether the sparse connectivity dynamically changes or not during training, sparse training usually can be divided into static sparse training (SST) and dynamic sparse training (DST).

**Static sparse training** represents a class of methods that train initial sparse neural networks with a fixed sparse connectivity pattern throughout training. While the sparse connectivity is static, the choices of the particular layer-wise sparsity (i.e., sparsity level of every single layer) can be diverse. The most naive approach is sparsifying each layer uniformly, i.e., uniform sparsity [Gale et al., 2019]. Mocanu et al. [2016] proposed a non-uniform sparsity method that can be applied in Restricted Boltzmann Machines (RBMs) and achieves better performance than dense RBMs. Some works explore the expander graph to train sparse CNNs and show comparable performance against the corresponding dense CNNs [Kepner and Robinett, 2019]. Inspired by the graph theory, *Erdős-Rényi* (ER) [Mocanu et al., 2018] and its CNNs variant *Erdős-Rényi-Kernel* (ERK) [Evci et al., 2020] allocates lower sparsity to smaller layers, avoiding the layer collapse problem [Tanaka et al., 2020] and achieving stronger results than the uniform sparsity in general.

**Dynamic sparse training**, namely, trains initial sparse neural networks while dynamically adjusting the sparse connectivity pattern during training. DST was first introduced in Sparse Evolutionary Training (SET) [Mocanu et al., 2018] which initializes the sparse connectivity with a ER topology and periodically explores the parameter space via a prune-and-grow scheme during training. Following SET, weights redistribution is introduced to search for better layer-wise sparsity ratios while training [Mostafa and Wang, 2019, Dettmers and Zettlemoyer, 2019]. The mainly-used pruning criterion of existing DST methods is magnitude pruning. The criterion used for weight regrowing varies from method to method. Gradient-based regrowth e.g., momentum [Dettmers and Zettlemoyer, 2019] and gradient [Evci et al., 2020], shows strong results in image classification,

whereas random regrowth outperforms the former in language modeling [Dietrich et al., 2021]. Follow-up works improve the accuracy by relaxing the constrained memory footprint [Yuan et al., 2021, Liu et al., 2021b]. Very recently, Liu et al. [2021a] proposed an efficient ensemble framework for sparse training– FreeTickets. By directly ensembling the predictions of individual subnetworks, FreeTickets surpass the generalization performance of the naive dense ensemble. Nevertheless, FreeTickets requires extending the training time to obtain multiple cheap subnetworks and performing multiple forward passes for inference, contrary to our pursuit of efficient training.

## 2.2 WEIGHT AVERAGING

Computing the convex combination of model weights usually leads to better robust performance Zhang et al. [2019], Neyshabur et al. [2020], Wortsman et al. [2022]. SWA [Izmailov et al., 2018] average weights along the same optimization trajectory with one single run. Neyshabur et al. [2020], in contrast, merge models that start with the same initialization but are optimized independently. Similarly, Wortsman et al. [2022] average models across many independent runs with various hyperparameters. Different from these prior works that only study on dense networks, we explore for the first time how to produce and combine multiple *sparse sub-networks* into a stronger one while considering the importance of the connectivities.

## 3 METHODOLOGY

In this section, we introduce a new approach for sparse training, which could combines the benefits of multiple cheap tickets, without extra training time and multiple forward passes for inference[Garipov et al., 2018, Liu et al., 2021a]. We first introduce the basic training scheme of sparse training in Section 3.1 and then describe our proposed Sup-tickets approach in detail in Section 3.2.

### 3.1 PRIOR SPARSE TRAINING ART

Following Liu et al. [2021c,a], we denote a sparse neural network as $f(x; \theta_s)$. $\theta_s$ refers to a subset of the full network parameters $\theta$ at a sparsity level of $(1 - \frac{\|\theta_s\|_0}{\|\theta\|_0})$, where $\|\cdot\|_0$ is the $\ell_0$-norm. Sparse training typically initializes the network in a random fashion where the connections between two adjacent layers are sparsely and randomly connected, based on a pre-defined uniform or non-uniform layer-wise sparsity ratio[2]. In the i.i.d. classification setting with data $\{(x_i, y_i)\}_{i=1}^N$, the goal of sparse training is to solve the following optimization problem:

---

[2]See Liu et al. [2022] for the most common types of sparse initialization.

$\hat{\theta}_s = \arg\min_{\theta_s} \sum_{i=1}^N \mathcal{L}(f(x_i; \theta_s), y_i)$, where $\mathcal{L}$ is the loss function. SST keeps the sparse connectivity of the sparse network fixed after initialization. DST, on the other hand, dynamically adjusts the sparse connectivity via parameter exploration during training while sticking to a fixed sparsity level. The most widely used method for parameter exploration is the prune-and-grow scheme, i.e., pruning $p\%$ the least important parameters from the current subnetwork followed by a fraction $p\%$ of weight growing. Formally, the parameter exploration can be written as the following two steps:

$$\theta_s' = \Psi(\theta_s, p), \tag{1}$$

$$\theta_s = \theta_s' \cup \Phi(\theta_{i \notin \theta_s'}, p) \tag{2}$$

where $\Psi$ and $\Phi$ are the specific pruning and growing criterion respectively. The choices of $\Psi$ and $\Phi$ differ from sparse training method to another. Besides the sparse structures, in the most sparse training literature [Dettmers and Zettlemoyer, 2019, Evci et al., 2020, Mostafa and Wang, 2019, Liu et al., 2021b], it is usually a safe choice to keep the other training configurations, such as optimizers, hyperparameters, and learning rate schedules, the same as the normal dense training. At the end of the training, sparse training can converge to a well-performing sparse subnetwork whose memory requirements, training, and inference FLOPs are only a fraction of the dense training.

### 3.2 SUP-TICKETS

Existing sparse training methods allocate all the limited resources to find the best sparse neural network possible. While low-loss subnetworks widely exist in the loss landscape of sparse neural network optimization [Liu et al., 2020], no prior works have ever explored how to find and leverage these handy cheap tickets to boost the performance of sparse training without extending training steps. In this section, we present Sub-tickets to close this research gap, as illustrated in Figure 1.

To achieve the above-mentioned ultimate goal, we need to satisfy the following two desiderata in one sparse-to-sparse training run:

1. **Creating cheap tickets**: Creating multiple cheap but well-performing subnetworks with one single run under a regular training time. We name such efficiently produced subnetworks as "cheap tickets".

2. **Superposing tickets**: Superposing these subnetworks into one subnetwork at the same sparsity to avoid performing multiple forward passes for the prediction. We term the "ultimate ticket" as the final subnetwork used for inference.

These two desiderata strictly follow the sparsity constraint of sparse training and thus maintain the training/inference

**Algorithm 1** Sup-tickets

**Require:** Network $f(\boldsymbol{x}; \boldsymbol{\theta})$, superposed subnetwork $\widetilde{\boldsymbol{\theta}}_s$, target sparsity $S$, training time $T$, cycle length $C$, learning rate $\alpha$, pruning criterion $\Psi$, growing criterion $\Phi$, pruning rate for parameter exploration $p$.

  1: $f(\boldsymbol{x}; \boldsymbol{\theta}_s) \leftarrow f(\boldsymbol{x}; \boldsymbol{\theta}; S)$                                          ▷*Sparsely initialize the network*
  2: **for** $i \leftarrow 1$ **to** $T$ **do**
  3:     **if** $i \leq 90\%T$ **then**                       ▷*Normal sparse training for the first 90% of T*
  4:         $f(\boldsymbol{x}; \boldsymbol{\theta}_s) \leftarrow SparseTraining(f(\boldsymbol{x}; \boldsymbol{\theta}_s))$
  5:     **else**                            ▷*Creating and superposing cheap tickets in the last 10% of T*
  6:         $\alpha \leftarrow \alpha(i)$                         ▷*Calculate the cyclical learning rate using Eq. 3*
  7:         $f(\boldsymbol{x}; \boldsymbol{\theta}_s) \leftarrow SparseTraining(f(\boldsymbol{x}; \boldsymbol{\theta}_s); \alpha)$
  8:         **if** $\mathrm{mod}(i - 90\%T, C) = 0$ **then**
  9:             $t \leftarrow (i - 90\%T)/C$               ▷*Number of the created cheap tickets*
10:             $\widetilde{\boldsymbol{\theta}}_s^t \leftarrow \frac{(t-1)\widetilde{\boldsymbol{\theta}}_s^{t-1} + \boldsymbol{\theta}_s^t}{t}$           ▷ *Ticket superposing using Eq. 4*
11:             $\widetilde{\boldsymbol{\theta}}_s^t \leftarrow MagnitudePruning(\widetilde{\boldsymbol{\theta}}_s^t)$     ▷*Prune the superposed ticket to the target sparsity S*
12:             $\boldsymbol{\theta}_s' \leftarrow \Psi(\boldsymbol{\theta}_s, p)$            ▷*Parameter exploration using Eq. 1 and Eq. 2*
13:             $\boldsymbol{\theta}_s \leftarrow \boldsymbol{\theta}_s' \cup \Phi(\boldsymbol{\theta}_{i \notin \boldsymbol{\theta}_{s'}}, p)$
14:         **end if**
15:     **end if**
16: **end for**
17: **Return** $\widetilde{\boldsymbol{\theta}}_s$                                        ▷*The ultimate ticket for test*

---

efficiency of sparse training.

### 3.2.1 Creating Cheap Tickets

During the last 10% of the training time, we cyclically explore the current sparse connectivity and restart the learning rate to visit multiple low-loss sub-space basins. More concretely, in each cycle, we first significantly change the connectivity of the current subnetwork by performing the parameter exploration once with Eq. 1 & 2. For simplicity, we inherit the pruning and growing methods used in the sparse training methods that Sup-tickets combines with. After parameter exploration, we leverage the cyclical learning rate to force the current subnetwork to escape the local minima. Inspired by Garipov et al. [2018], Izmailov et al. [2018], we adopt the learning rate schedule scheme as:

$$\alpha(i) = \begin{cases} (1 - 2t(i))\alpha_1 + 2t(i)\alpha_2 & 0 < t(i) \leq \frac{1}{2} \\ (2 - 2t(i))\alpha_2 + (2t(i) - 1)\alpha_1 & \frac{1}{2} < t(i) \leq 1 \end{cases} \quad (3)$$

where $\alpha(i)$ is the cyclical learning rate ranging from $\alpha_1$ to $\alpha_2$; $i$ is the training iteration for one mini-batch data; $t(i) = \frac{1}{C}(\mathrm{mod}(i - 1, C) + 1)$; $C$ is the cycle length. We modify the cyclical learning rate schedule used in SWA [Izmailov et al., 2018] to prevent the aggressive rise of the learning rate. Specifically, we adopt the triangle-like schedule as shown in Figure 2-bottom. In such a way, the learning rate could seamlessly transition from the normal training stage to the superposing stage. At the end of each cycle, we can obtain one cheap ticket from the current basin with diverse and meaningful representation.

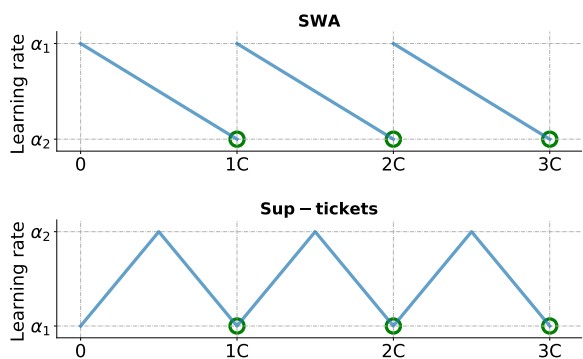

Figure 2: **Top:** cyclical learning rate schedule of Garipov et al. [2018]. **Bottom:** cyclical learning rate schedule of Sup-tickets. Cheap tickets are collected at the end of each learning rate schedule cycle (green circles in the figure).

The combination of cyclical learning rate schedule and parameter exploration is also used in FreeTickets [Liu et al., 2021a], but we have several distinctions to make it compiled with the requirements of sparse training. The cycle duration of FreeTickets is set as 100 epochs to guarantee the consistent strong performance of each subnetwork as they try to achieve comparable performance with the dense ensemble. However, such a long duration of cycle conflicts with the goal of sparse training. In particular, we reduce the cycle duration to 2 epochs for ImageNet, 8 epochs for CIFAR-10/100 and only use the final 10% of the training time to generate cheap tickets. In this case, the overall training time is the same as training a single sparse network.

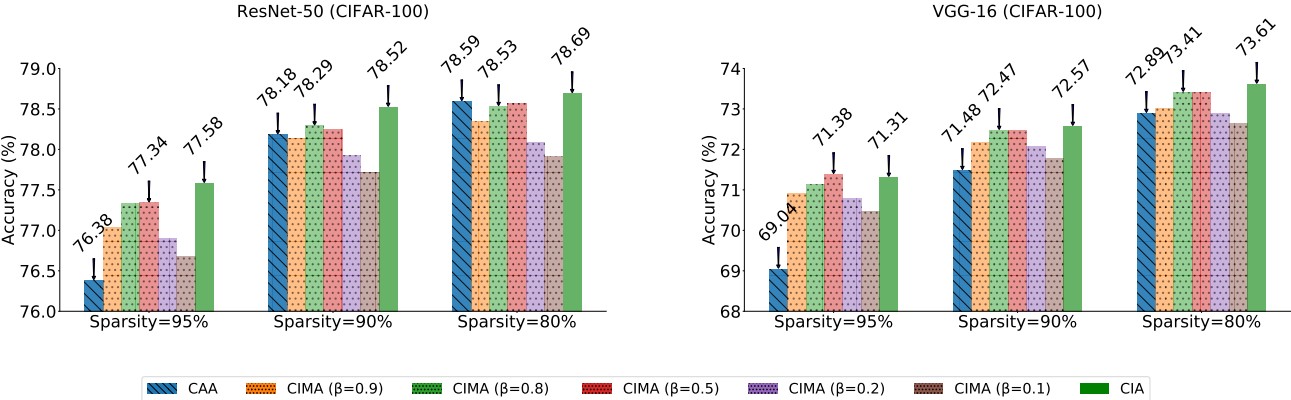

Figure 3: Comparisons of various averaging methods. We combine CIA, CAA, and CIMA with RigL and report the test accuracy of the ultimate tickets. For CIMA, we vary the exponential decay rates $\beta \in [0.9, 0.8, 0.5, 0.2, 0.1]$.

### 3.2.2 Superposing Tickets

Superposing multiple sparse networks is more complex than superposing multiple dense networks [Cheung et al., 2019, Izmailov et al., 2018]. Naively selecting all the weights that are activated in all cheap tickets will significantly increase the parameter count, as different subnetworks have different connectivities. To solve this task, we propose to perform weight averaging followed by weight pruning. More concretely, assuming we collect M cheap tickets $\{\theta_s^1, \theta_s^2, ..., \theta_s^M\}$ at the end of training, we consider the following three ways to average them.

**Connection Independent Averaging (CIA).** The ultimate subnetwork averaged by CIA is given as: $\widetilde{\theta}_{s'} = \frac{1}{M} \sum_{i=1}^M \theta_s^i$, where M is the total number of cheap tickets. CIA simply averages weights across all the cheap tickets without considering whether the connection is activated or not in each cheap ticket. CIA tends to preserve the connections that are activated in the majority of the cheap tickets whereas the ones that are occasionally activated in one or two cheap tickets are likely to have small magnitude after averaging by M, unless they have extremely large values.

**Connection Aware Averaging (CAA).** The ultimate subnetwork averaged by CAA is given as: $\widetilde{\theta}_s = \frac{1}{N(k,j)} \sum_{i=1}^M \theta_s^i$, where $N(k,j)$ is the number of times the connection $\theta(k,j)$ is activated across all the cheap tickets; $k$ is the $k^{th}$ neuron in the previous layer and $j$ is the $j^{th}$ neuron in this layer. Thus, we have $N(k,j) \leq M$. Compared with CIA, CAA pays more attention to the occasionally activated connections that are only existing in the minority of cheap tickets.

**Connection Independent Moving Averaging (CIMA).** Motivated by the widely-used moving average technique [Kingma and Ba, 2014, Karras et al., 2017], we sequentially apply the popular moving averages over the cheap tickets obtained at each cycle. The averaged sub-

network over the first $t$ cheap tickets is given as: $\widetilde{\theta}_s^t = \beta\widetilde{\theta}_s^{t-1} + (1-\beta)\theta_s^t$. $\beta$ controls the exponential decay rates. Larger $\beta$ will put more emphasis on the cheap tickets collected in the early time.

Note that the sparsity of the averaged subnetwork is likely larger than the target sparsity level. To maintain the same sparsity as the original subnetwork, we utilize magnitude weight pruning to remove the weights with the smallest magnitude after every averaging step.

### 3.3 MEMORY AND COMPUTATION OVERHEAD

Instead of saving M cheap tickets and average them, we apply a similar operation as CIMA to save the extra memory required by CIA and CAA during training. The averaged subnetwork over the first $t$ cheap tickets is given as:

$$\widetilde{\theta}_s^t = \frac{(t-1) \cdot \widetilde{\theta}_s^{t-1} + \theta_s^t}{t} \tag{4}$$

This operation allows us to accomplish the average operation by maintaining only one extra copy of the averaged weights, instead of saving M subnetworks.

Moreover, as we mentioned, we use the final 10% of the training time to create cheap tickets, and thus the training time of Sub-tickets is the same as the standard sparse training. Since we only need to perform Eq. 4 for (M − 1) times, the extra computation cost of averaging is negligible compared with the total training costs. Overall, we can conclude that the training cost of Sub-tickets is approximately the same as training a single sparse network.

## 4 EXPERIMENTS

Sub-tickets is a universal idea that can be straightforwardly applied to any types of sparse training methods. To ver-

ify the effectiveness of Sup-tickets, we apply it to various sparse training methods, including 3 DST methods: SET, RigL [Evci et al., 2020], and GraNet [Liu et al., 2021b]; one SST method: ERK [Evci et al., 2020]; and one pruning at initialization approach: SNIP [Lee et al., 2018].

## 4.1 EXPERIMENTAL SETUPS

The experiments are conducted across various architectures on three popular datasets CIFAR-10/100 and ImageNet. For CIFAR-10/100, we choose models VGG-16 [Simonyan and Zisserman, 2014], Wide ResNet28-10 [Zagoruyko and Komodakis, 2016] and ResNet-50 [He et al., 2016]. The models are trained for 250 epochs, optimized by momentum SGD with a learning rate of 0.1, which decayed by 10x at the half and three-quarters of the training stage. The cycle length is chosen as 8 epochs, so that we can obtain 3 cheap tickets in 24 epochs. The model used for ImageNet is ResNet-50, which is trained for 100 epochs, optimized by momentum SGD with a learning rate of 0.1 decaying by 10x at 30, 60, and 85 epoch. The cycle length of ImageNet is 2 epochs, so we obtain 4 cheap tickets in the last 8 epochs. The implementation details are reported in Appendix E.

## 4.2 COMPARISONS AMONG CIA, CAA, AND CIMA

We first conduct a comparison among CIA, CAA, and CIMA on CIFAR-100 and report the results in Figure 3. We can see that CIA consistently outperforms the other two methods at various sparsity levels. CAA is the worst-performing method, especially at the extreme sparsity 95%. With tuned $\beta = 0.8$, CIMA can approach the performance achieved by CIA. The better performance achieved by CIA over CAA indicates that the occasionally activated connections are likely unimportant. CIA pays more attention to the connections that exist in the majority of the cheap tickets, which can eliminate the unimportant connections that are activated occasionally. Therefore, due to the superior performance consistently achieved by CIA, we choose CIA as our averaging method in the following sections.

## 4.3 EVALUATION OF SUP-TICKETS

**CIFAR-10/100.** In this section, we provide an experimental comparison of Sup-tickets to a variety of sparse training techniques. The results of CIFAR-10/100 with VGG-16 and ResNet-50 are shown in Table 1 & 2 respectively, and the results of Wide ResNet28-10 are shared in Appendix A due to the limited space. Overall, we clearly see that our approach could benefit sparse training across all studied architectures. Simple as it looks, Sup-tickets improves the performance of various dynamic sparse training methods in 63 out of 66 cases. It seems Sup-tickets performs better with

VGG-16 than the other two architectures, with up to 0.5% and 1.08% accuracy increase on CIFAR-10 and CIFAR-100, respectively. We also find that the performance improvement on CIFAR-100 is larger than the one on CIFAR-10, which makes sense since CIFAR-100 is less saturated and thus has a larger improvement space. More importantly, our approach combined with the state-of-the-art DST method – GraNet, outperforms the dense networks with only about 5% at most 10% parameters with all architectures, as reported in Table 4. All these results highlight that Sup-tickets is a strong and universal performance booster for sparse training.

**ImageNet.** For ImageNet, we apply Sup-tickets to RigL and GraNet and compare them with the existing sparse training methods. The results are reported the in Table 3. Again, we improve the performance of GraNet and RigL at both 80% sparsity and 90% sparsity without an extra parameter budget. Especially on RigL, our approach improves the test accuracy by 0.9% and 1.0% at sparsity 80% and 90%, respectively. Besides, we compare the Sup-tickets with the naive deep ensemble method and show the results in Appendix H.

Examining the results, we note that Sup-tickets improve both SST and DST in all settings with a small operation modification of those algorithms. In all settings, a large array of other techniques are outperformed.

## 5 EXTENSIVE ANALYSIS

**Cyclical Length.** Here, we study how the cyclical length $C$ affects the Sup-tickets' performances. For all experiments, we still take the last 10% of the training time for the generation of the cheap tickets, while altering the cyclical length as 2, 4, 8, and 12 epochs. The cheap ticket count then varies accordingly. The results are shown in Table 5. In general, the intermediate lengths (i.e., $C = 4$ or $C = 8$) tend to achieve better accuracy than the extreme small or large lengths (i.e., $C = 2$ or $C = 12$). The results are expected since small lengths can not guarantee the high quality (high accuracy) of each cheap ticket, whereas large lengths naturally decrease the number of the collected tickets. Consequently, we use $C = 8$ as the default setting in the main experiment section 4.3.

**Number of Cheap Tickets.** To study the effect of the cheap ticket count on ultimate ticket's performance, we alter the cheap ticket count with 2, 4, and 7, and fix the cyclical length as 8 epochs. The overall training time is set as 250 epochs. Under this setting, the time used for ticket generation is not fixed as 10%, but it changes according to the cheap ticket count. We report the results in Figure 4-left. It could be seen that our approach achieves the best performance under four tickets, not the largest nor the smallest ticket count, apparently since creating too many cheap tickets will reduce the time of the normal sparse training phase, and thus yielding cheap tickets with

Table 1: Test accuracy (%) of sparse VGG-16 on CIFAR-10/100. All the results are averaged from three random runs. In each setting, the best results are marked in bold.

| Dataset | CIFAR-10 | | | CIFAR-100 | | |
|---|---|---|---|---|---|---|
| **VGG-16** (Dense) | 93.91±0.26 | - | - | 73.61±0.45 | - | - |
| Sparsity | 95% | 90% | 80% | 95% | 90% | 80% |
| SET [Mocanu et al., 2018] | 92.96±0.18 | 93.54±0.23 | 93.56±0.04 | 70.10±0.33 | 71.50±0.23 | 72.38±0.08 |
| SET+Sup-tickets (ours) | **93.22±0.09** | **93.63±0.05** | **93.80±0.13** | **71.18±0.29** | **71.99±0.27** | **73.02±0.32** |
| RigL [Evci et al., 2020] | 92.70±0.08 | 93.48±0.16 | 93.60±0.14 | 70.65±0.16 | 72.20±0.09 | 72.63±0.23 |
| RigL+Sup-tickets (ours) | **93.20±0.13** | **93.81±0.11** | **93.85±0.25** | **71.31±0.21** | **72.57±0.29** | **73.61±0.11** |
| GraNet [Liu et al., 2021b] | 93.87±0.19 | 93.83±0.30 | 93.77±0.18 | 72.91±0.39 | 73.48±0.17 | 73.36±0.14 |
| GraNet+Sup-tickets (ours) | **94.10±0.06** | **94.13±0.12** | **94.24±0.05** | **73.61±0.24** | **73.87±0.26** | **73.95±0.30** |

Table 2: Test accuracy (%) of sparse ResNet-50 on CIFAR-10/100. All the results are averaged from three runs. In each setting, the best results are marked in bold.

| Dataset | CIFAR-10 | | | CIFAR-100 | | |
|---|---|---|---|---|---|---|
| **ResNet-50** (Dense) | 94.88±0.11 | - | - | 78.00±0.40 | - | - |
| Sparsity | 95% | 90% | 80% | 95% | 90% | 80% |
| SNIP [Lee et al., 2018] | 94.01±0.28 | 94.81±0.36 | 94.91±0.16 | 41.25±1.10 | 68.79±1.16 | 75.29±1.28 |
| SNIP+Sup-tickets (ours) | **94.33±0.09** | **95.05±0.22** | **95.21±0.09** | **65.56±1.15** | **76.34±0.27** | **77.43±0.53** |
| ERK [Evci et al., 2020] | 93.44±0.22 | 94.41±0.13 | 94.85±0.21 | 74.49±0.30 | 76.36±0.22 | 77.41±0.08 |
| ERK+Sup-tickets (ours) | **93.92±0.04** | **94.80±0.06** | **95.11±0.27** | **75.75±0.28** | **76.82±0.08** | **77.85±0.42** |
| SET [Mocanu et al., 2018] | 94.49±0.11 | 94.73±0.27 | 94.74±0.17 | 76.59±0.54 | 77.79±0.27 | **78.45±0.50** |
| SET+Sup-tickets (ours) | **94.81±0.05** | **94.87±0.03** | **94.90±0.27** | **76.68±0.38** | **77.89±0.45** | 78.35±0.18 |
| RigL [Evci et al., 2020] | 94.59±0.19 | 94.70±0.17 | 94.70±0.07 | 76.96±0.39 | 77.95±0.36 | 78.19±0.51 |
| RigL+Sup-tickets (ours) | **94.65±0.11** | **94.82±0.13** | **94.81±0.15** | **77.58±0.47** | **78.52±0.39** | **78.69±0.30** |
| GraNet [Liu et al., 2021b] | 94.70±0.23 | 94.95±0.09 | 94.86±0.24 | 77.47±0.22 | 78.25±0.51 | 78.80±0.46 |
| GraNet+Sup-tickets (ours) | **94.89±0.15** | **95.08±0.08** | **94.94±0.03** | **77.70±0.47** | **78.37±0.53** | **78.95±0.33** |

Table 3: Test accuracy (%) of sparse ResNet-50 on ImageNet. The training FLOPs of sparse training methods are normalized with the FLOPs used to train a dense dense model. In each setting, the best results are marked in bold.

| Method | Top-1 Accuracy | FLOPs (Train) | FLOPs (Test) | TOP-1 Accuracy | FLOPs (Train) | FLOPs (Test) |
|---|---|---|---|---|---|---|
| **ResNet-50** (Dense) | 76.8±0.09 | 1x (3.2e18) | 1x (8.2e9) | 76.8±0.09 | 1x (3.2e18) | 1x (8.2e9) |
| Sparsity | | 80% | | | 90% | |
| Static sparse training (ERK) | 72.1±0.04 | 0.42× | 0.42× | 67.7±0.12 | 0.24× | 0.24× |
| Small-Dense | 72.1±0.06 | 0.23× | 0.23× | 67.2±0.12 | 0.10× | 0.10× |
| SNIP [Lee et al., 2018] | 72.0±0.06 | 0.23× | 0.23× | 67.2±0.12 | 0.10× | 0.10× |
| SET [Mocanu et al., 2018] | 72.9±0.39 | 0.23× | 0.23× | 69.6±0.23 | 0.10× | 0.10× |
| DSR [Mostafa and Wang, 2019] | 73.3 | 0.40× | 0.40× | 71.6 | 0.30× | 0.30× |
| SNFS [Dettmers and Zettlemoyer, 2019] | 75.2±0.11 | 0.61× | 0.42× | 72.9±0.06 | 0.50× | 0.24× |
| RigL [Evci et al., 2020] | 75.1±0.05 | 0.42× | 0.42× | 73.0±0.04 | 0.25× | 0.24× |
| RigL+Sup-tickets (ours) | **76.0** | 0.42× | 0.42× | **74.0** | 0.25× | 0.24× |
| GraNet [Liu et al., 2021b] | 75.9 | 0.37× | 0.35× | 74.4 | 0.25× | 0.20× |
| GraNet+Sup-tickets (ours) | **76.2** | 0.37× | 0.35× | **74.6** | 0.25× | 0.20× |

Table 4: Performance comparison between GraNet+Sup-tickets and dense network. Results that are better than the corresponding dense networks are marked in bold. WRN28-10 refers to Wide ResNet28-10. GraNet+Sup-tickets outperforms dense network in most cases.

| Dataset | Network | Dense | GraNet+Sup-tickets | | |
|---|---|---|---|---|---|
| | | | 95% sparsity | 90% sparsity | 80% sparsity |
| CIFAR-10 | VGG-16 | 93.91±0.26 | **94.10±0.06** | **94.13±0.12** | **94.24±0.05** |
| | ResNet-50 | 94.88±0.11 | **94.89±0.15** | **95.08±0.08** | **94.94±0.03** |
| | WRN28-10 | 96.00±0.13 | **96.03±0.11** | **96.13±0.07** | **96.08±0.04** |
| CIFAR-100 | VGG-16 | 73.61±0.45 | 73.61±0.24 | **73.87±0.26** | **73.95±0.30** |
| | ResNet-50 | 78.00±0.40 | 77.70±0.47 | **78.37±0.53** | **78.95±0.33** |
| | WRN28-10 | 81.09±0.19 | 80.65±0.06 | **81.20±0.09** | **81.42±0.18** |

Table 5: Test accuracy (%) on CIFAR-100 of Sup-tickets combined with RigL under different cyclical lengths. The best results are marked in bold.

| Cyclical | Pruning ratio | | |
|---|---|---|---|
| length (epochs) | 95% | 90% | 80% |
| VGG-16 | | | |
| C=2 | 71.35±0.14 | 72.89±0.41 | **73.65±0.20** |
| C=4 | **71.42±0.19** | **73.00±0.20** | 73.62±0.40 |
| C=8 | 71.31±0.21 | 72.57±0.29 | 73.61±0.11 |
| C=12 | 71.27±0.06 | 72.69±0.43 | 73.45±0.06 |
| ResNet-50 | | | |
| C=2 | **77.58±0.22** | 78.48±0.45 | 78.50±0.32 |
| C=4 | 77.33±0.26 | **78.52±0.36** | 78.62±0.34 |
| C=8 | **77.58±0.47** | **78.52±0.39** | **78.69±0.30** |
| C=12 | 77.17±0.42 | 78.39±0.43 | 78.48±0.38 |

poor performance. We further prove this in Figure 4-right. On the other hand, 2 cheap tickets are too few to boost the performance. Figure 4 also illustrates the effectiveness of Sup-tickets, where the superposed subnetworks outperform the individual subnetworks by a large margin.

The fixed training time constraint is important to enable comparisons among various sparse training methods since training efficiency is one of the main contributions of sparse training. It is natural to evaluate whether Sup-tickets can lead to continuous improvement when we remove this constraint. To evaluate this, we simply extend the overall training time to yield more cheap tickets. The results are reported in Appendix B. We can see that the performance of Sup-tickets continuously improves as the number of tickets increases.

**Diversity Analysis.** We report the diversity of the different subnetworks we obtained during training using KL divergence and prediction disagreement, which are widely used for deep ensembling Liu et al. [2021a], Fort et al. [2019]. We compare our methods against the traditional dense ensemble and two state-of-the-art efficient ensemble methods, including TreeNet [Lee et al., 2015] and BatchEnsemble [Wen et al., 2020], with Wide ResNet28-10 on CIFAR-10. The results are also in line with our intuition. We observe that the diversity of cheap tickets obtained by our method is lower than the traditional dense ensemble. This makes sense since

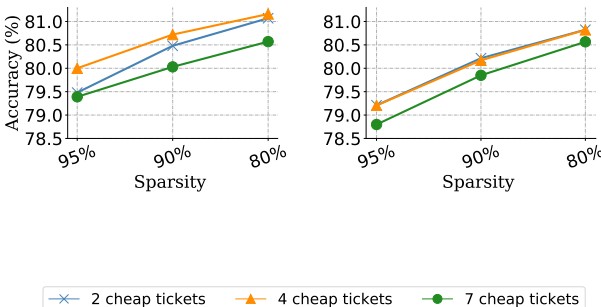

Figure 4: Impacts of the cheap tickets count. Experiments are conducted with Wide ResNet28-10 trained with RigL+Sup-tickets on CIFAR-100. **Left:** test accuracy of the ultimate tickets. **Reft:** the mean accuracy of the individual cheap tickets used to build the ultimate tickets.

networks of the traditional dense ensemble are obtained by different runs and should converge to different basins, whereas cheap tickets obtained by our methods are intended to be located in the same basin with relatively lower diversity. Nevertheless, our method still maintains a similar or even higher diversity than TreeNet and BatchEnsemble, verifying its effectiveness. The relatively low diversity ensures that our cheap tickets are located in the same wide and flat low loss region, which is actually crucial for the success of weight averaging, since too diverse networks could lead to very poor performance from the previous experiments Izmailov et al. [2018], Wortsman et al. [2021].

Table 6: Prediction disagreement and KL divergence among various ensemble methods.

| Methods | $d_{\mathrm{dis}}$ (↑) | $d_{\mathrm{KL}}$ (↑) |
|---|---|---|
| TreeNet Lee et al. [2015] | 0.010 | 0.010 |
| BatchEnsemble Wen et al. [2020] | 0.014 | 0.020 |
| SET+Sup-tickets (ours) | 0.015 | 0.015 |
| Rigl+Sup-tickets (ours) | 0.017 | 0.015 |
| Traditional Dense Ensemble | 0.032 | 0.086 |

**Comparison with Different Learning Rate Schedules.** We compare our method with two learning rate schedule baselines: the learning rate schedule used in FGE [Garipov et al., 2018] and the learning rate schedule used in SWA [Izmailov et al., 2018]. In all schedules, Sup-tickets are collected at the lowest learning rate stage, and we fixed the learning rate range of these schedules for a fair comparison. Below we report the results on CIFAR-100. All the results are averaged from 3 random runs. It could be seen that our method surpasses the other baselines in 5 out of 6 cases.

We adjust the learning rate schedule slightly so that the learning rate gradually rises to an increased but still small value (0.005) and then decays to the lowest value (0.001) in each cycle. Such a smooth schedule ensures that the new

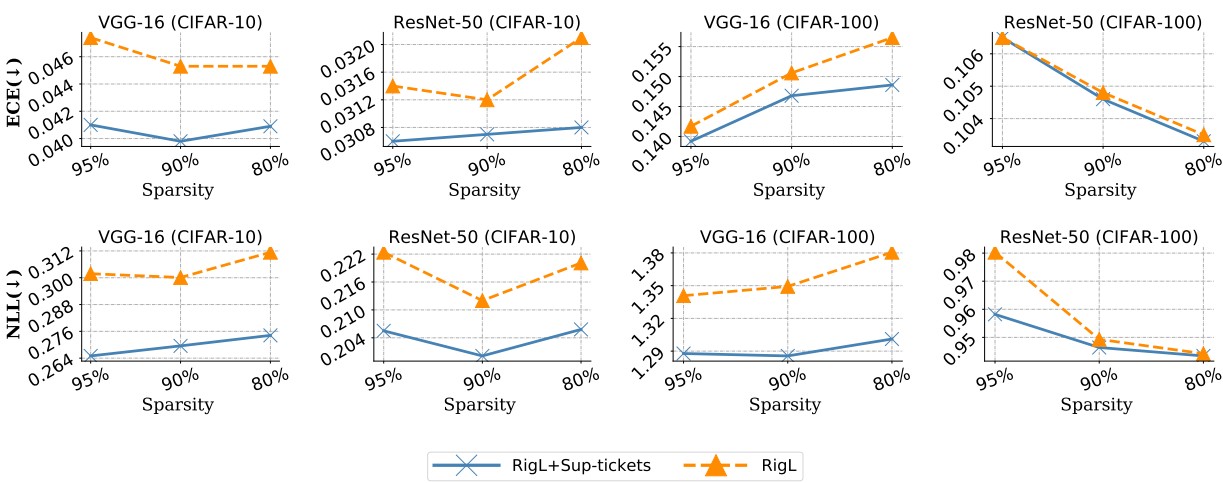

Figure 5: Comparison between RigL and RigL+Sup-tickets in terms of ECE and NLL.

Table 7: Effect of Various Different Learning Rate (LR) Schedules.

| LR schedule | Sparsity | | |
|---|---|---|---|
| Method | 95% | 90% | 80% |
| VGG-16 | | | |
| LR of FGE [Garipov et al., 2018] | 70.66±0.25 | 72.47±0.44 | 73.22±0.23 |
| LR of SWA [Izmailov et al., 2018] | 71.26±0.16 | **72.77±0.37** | 73.44±0.19 |
| Sup-ticket (Ours) | **71.31±0.21** | 72.57±0.29 | **73.61±0.11** |
| ResNet-50 | | | |
| LR of FGE [Garipov et al., 2018] | 77.30±0.67 | 78.20±0.53 | 78.35±0.35 |
| LR of SWA [Izmailov et al., 2018] | 77.30±0.36 | 78.39±0.38 | 78.48±0.35 |
| Sup-ticket (Ours) | **77.58±0.47** | **78.52±0.39** | **78.69±0.30** |

cheap tickets only bounce within the same basin instead of jumping out of it. To help us clarify this, we added extra experiments and report the results in Appendix D.

**Batch Normalization.** When there are batch normalization (BN) layers [Ioffe and Szegedy, 2015] in the model, traditional weight averaging approaches [Garipov et al., 2018, Izmailov et al., 2018] usually run one additional pass over the data to calculate the mean and standard deviation of these layers. Differently, we retrieve these statistics by simply averaging the mean and standard deviation of the BN layers in all cheap tickets without extra forward pass. To avoid extra memory occupation during implementation, similar to the weights averaging operation in Eq. 4, we calculate the superposed ticket's BN statistics $\widetilde{\boldsymbol{\theta}}_{\mathrm{bn}}^{\mathrm{t}}$ across the first $t$ cheap tickets using $\frac{(t-1)\cdot\widetilde{\boldsymbol{\theta}}_{\mathrm{bn}}^{\mathrm{t-1}}+\boldsymbol{\theta}_{\mathrm{bn}}^{\mathrm{t}}}{t}$, where $\boldsymbol{\theta}_{\mathrm{bn}}^{\mathrm{t}}$ is the mean and standard deviation from $t^{th}$ cheap ticket's BN layers. The comparison between test accuracy under these two strategies is reported in Appendix F.

**Uncertainty Estimation.** In the security-critical scenarios, e.g., self-driving, medical treatment, classifiers should not only be accurate but also indicate when they are likely to be incorrect [Guo et al., 2017]. We further evaluate the per-

formance of our approach on uncertainty estimation. We choose two widely-used metrics, expected calibration error (ECE) [Guo et al., 2017] and negative log-likelihood (NLL) [Quinonero-Candela et al., 2005] to enable uncertainty comparisons among different methods. We apply Sup-tickets to RigL and compare it with the vanilla RigL in Figure 5. As observed, in addition to the improvement of accuracy, Sup-tickets also achieves stronger uncertainty estimation performance over RigL, and such improvement can likely generalize to other sparse training methods.

## 6 CONCLUSION

In this paper, we presented a novel sparse training approach, Sup-tickets, which effectively produces many cheap subnetworks (tickets) during training and superposes them into one stronger ultimate subnetwork. Sup-tickets is easily combined with existing techniques, agnostic to model architectures, datasets, and is able to boost the sparse training performance with only a negligible amount of extra FLOPs. Across various scenarios, consistent performance improvement is obtained by Sup-tickets in terms of accuracy as well as uncertainty estimation, under the same training time used by the standard sparse training methods. It is impressive to see that sup-tickets outperforms the corresponding dense networks on CIFAR-10/100 even in extremely sparse situations when collaborating with GraNet.

There are many potential directions to be explored in the future. For example, even if Sup-tickets enable sparse neural networks to match or outperform their dense counterparts in terms of test accuracy, do they learn the same representation as the latter learn? Besides, we hope the superior performance achieved by Sup-tickets could inspire more researchers to invest in developing hardware accelerators that have better support for sparse training.

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
