# OpenReview forum: "Superposing Many Tickets into One: A Performance Booster for Sparse Neural Network Training"
_auai.org/UAI/2022/Conference — UAI 2022 Poster_

### Official Review · Reviewer_kbVe · 2022-04-11

**Q2(1) Originality/Novelty:** 3
**Q2(2) Significance/Impact:** 3
**Q2(3) Correctness/Technical Quality:** 3
**Q2(6) Clarity Of Writing:** 4
**Q6 Overall Score:** 7
**Q8 Confidence In Your Score:** 4

**Q1 Summary And Contributions:**

In this paper the authors provide a method for sparse training that efficiently finds very sparse sub-networks which still performs well. These sub-networks, also called cheap tickets, are then combined to create a better performing model that still retains the sparsity benefits.

The authors then provide empirical evidence that their approach performs well on several classification tasks.

**Q2 Assessment Of The Paper:**

More detailed information regarding each of these aspects is given below:

**Q2(4) Quality Of Experiments (Optional):**

3: Good: The experimental evaluation is adequate, and the results convincingly support the main claims.

**Q2(5) Reproducibility:**

3: Good: Key resources (e.g., proofs, code, data) are available and key details (e.g., proofs, experimental setup) are sufficiently well-described for competent researchers to confidently reproduce the main results.

**Q3 Main Strengths:**

The provided approach manages to sparsely train neural networks keeping the computational costs low while providing highly accurate models.

The different approaches to merge the cheap tickets are simple, interesting and effective. The empirical results confirm that they work well without increasing the training costs.

The author compares to other related-works showing improvements across the board.

**Q4 Main Weakness:**

The authors focus only on classification tasks, raising the question of how general this sparse training approach is. Would it also perform well on other architectures, e.g., GANs? VAEs?, etc.

Considering tables 1 and 2, we see that the values are although marginally better still impressive given the sparsity. However, are they statistically significant? This is not mentioned. This is important as some of the other approaches have a similar computational complexity.

**Q5 Detailed Comments To The Authors:**

In general I find the paper interesting and useful. It is also easy to read and technically sound.

The issues mentioned in Q4 are still relevant, especially the issues around the significance which would make the reader more confident about the results.

There are other minor questions that could also be interesting. Do you use regularization? I do not remember any discussion about that, nor whether you use dropout in your approaches. Would that help or hurt in this highly sparse scenario? A small comment on that or references would be helpful for reproducibility too.

I found the examination of the number of tickets, the cyclical length and the comparisons in terms of ECE and NLL interesting and carefully thought out.


**Q7 Justification For Your Score:**

In general I find this is a good paper, covering the different aspects presented in the approach very carefully and easy to read. My only serious comments would be regarding the significance of your results.

**Q9 Complying With Reviewing Instructions:**

1: Yes.

---

### Official Review · Reviewer_q3AE · 2022-04-13

**Q2(1) Originality/Novelty:** 2
**Q2(2) Significance/Impact:** 2
**Q2(3) Correctness/Technical Quality:** 3
**Q2(6) Clarity Of Writing:** 4
**Q6 Overall Score:** 6
**Q8 Confidence In Your Score:** 4

**Q1 Summary And Contributions:**

The paper proposes an approach to enable more efficient training of sparse neural networks. The main idea is to average parameters of multiple instances of trained sparse networks, which they call cheap tickets or sub tickets or sub networks. Thus, instead of spending N iterations to train one sparse network until convergence, we spend the same amount of compute to train K sparse networks (each for N/K iterations), but achieve superior performance by combining their parameters.



**Q2 Assessment Of The Paper:**

More detailed information regarding each of these aspects is given below:

**Q2(4) Quality Of Experiments (Optional):**

3: Good: The experimental evaluation is adequate, and the results convincingly support the main claims.

**Q2(5) Reproducibility:**

3: Good: Key resources (e.g., proofs, code, data) are available and key details (e.g., proofs, experimental setup) are sufficiently well-described for competent researchers to confidently reproduce the main results.

**Q3 Main Strengths:**

- The core of this idea of using an approach to have the advantage of an ensemble method, without the need to keep all subnetworks is interesting and can be very useful in practical scenarios.  Averaging the parameters of these cheap tickets, instead of using a standard ensemble method of averaging activations/outputs, helps to avoid the cost of maintaining and running multiple networks during inference.
- The paper touches upon an interesting challenge of how/when to collect the cheap tickets.
- The paper is well organised and easy/clear to read.
- The results are properly reported, over different trials and with respect to different metrics (compute time, accuracy, confidence calibration).
- The paper compares different strategies for averaging the parameters of the subnetworks and provide some intuitions about why/how they could lead to different outcomes.

**Q4 Main Weakness:**

- Lack of evidence and discussion about the similarity or dissimilarity of the cheap tickets collected with the proposed approach.
- Lack of discussion, theoretical or intuitive, that why averaging the networks at parameter level, in this particular case, should lead to well behaving models.
- While the main novelty of the paper, could be in the strategy used to collect the cheap tickets, this particular aspect of the proposed method is not compared to any baselines.
- The relation with prior work and the novelty of the contributions is not well established.

**Q5 Detailed Comments To The Authors:**


The methods consists of two main steps:
1. Training and collecting the subnetworks:  To do so, through the coarse of training sparse NNs (with) they apply a specific cyclic learning rate schedule is applied together with a parameter exploration technique.
2. Averaging parameters of the subnetworks: since we are in the sparse training regime, each subnetwork could converge to a different sparsity patterns, and the authors examine how different averaging strategies  impacts the results.


My take is, the main contribution/novelty of the paper is in the way they collect the subnetworks. Unfortunately, in the experiments, the proposed strategy for doing this is not compared with any baseline. E.g., using different learning rate scheduling, or simply sampling subnetworks randomly during a standard coarse of training.

Moreover, my main concern about the second step: We know from prior work, e.g., that computing convex combination of parameters of instances of NNs can lead to better/more robust models. However, this happens under specific conditions. E.g., if instances are close enough and in the same basin in the loss landscape.  I think it’s worth it, if the authors can intuitively or theoretically explain why this approach should work in this case.  Basically, my understanding from the paper is that the strategy for collecting the sub networks is designed such that these instances belong to different basins in the loss landscape. Hence, it’s not clear to me why we should expect the average of the parameters of these models to be a well behaving model? Taking this into account, and the fact that the results indicate some improvements, I am guessing that the assumption that these cheap tickets belong to different basins in the loss landscape, does not hold. Maybe the authors can analyse this, and provide some evidence for whether this is actually the case or not.

Additionally, while it’s really nice that the paper compares different strategies for averaging the parameters of the subnetworks. I wonder how the results would compare to the ensemble output of these subnetworks (averaging outputs/activations). On this note, to avoid dealing with issues related to averaging parameters of the networks, maybe distillation ( distilling from all sub-networks into one), can be used as an alternative strategy.



**Q7 Justification For Your Score:**

I have concerns about both novelty and correctness of the approach.
1. Whether averaging the parameters of subnetworks is a meaningful operation in the particular setup of this paper? The paper can have a much stronger point, if this is justified and the improvements in the results and discussion about why this should lead to any improvements are consistent.
2. The experiments don't support the fact that this particular strategy of collecting cheap tickets is leading to better results.


**Q9 Complying With Reviewing Instructions:**

1: Yes.

---

### Official Review · Reviewer_XxCS · 2022-04-16

**Q2(1) Originality/Novelty:** 3
**Q2(2) Significance/Impact:** 3
**Q2(3) Correctness/Technical Quality:** 3
**Q2(6) Clarity Of Writing:** 3
**Q6 Overall Score:** 7
**Q8 Confidence In Your Score:** 3

**Q1 Summary And Contributions:**

This paper focuses on learning sparse neural networks. Instead of learning a sparse network from scratch, the paper proposes to learn multiple low-loss sparse subnetworks (cheap tickets) and then superpose them together to get one stronger sparse network while maintaining the sparse degree. Experiments show the effectiveness on different datasets with different baseline models.

**Q2 Assessment Of The Paper:**

More detailed information regarding each of these aspects is given below:

**Q2(4) Quality Of Experiments (Optional):**

3: Good: The experimental evaluation is adequate, and the results convincingly support the main claims.

**Q2(5) Reproducibility:**

3: Good: Key resources (e.g., proofs, code, data) are available and key details (e.g., proofs, experimental setup) are sufficiently well-described for competent researchers to confidently reproduce the main results.

**Q3 Main Strengths:**

+ The paper is well motivated and well written. It is easy to follow. The claim is clear.
+ Extensive experiments are conducted to show the performances of the proposed method and study the effectiveness of different subcomponents, such as the averaging method for superposing operation.


**Q4 Main Weakness:**

- More detailed discussions are absent.
- It is still unclear why more cheap tickets cause worse performance shown in Figure 5. Why “creating too many cheap tickets will reduce the time of the normal sparse training phase”? What if enforcing not “reducing the time of the normal sparse training phase”? Could it boost the performance with more cheap tickets? This observation conflicts with some basic assumptions.
- How is the variance of the multiple cheap tickets? In case there may be high variance, does it influence the accuracy of the averaging-based method?


**Q5 Detailed Comments To The Authors:**

I put the details with main weakness.

**Q7 Justification For Your Score:**

The paper is well motivated and well written. It is easy to follow. Abundant experiments prove the main claims.

**Q9 Complying With Reviewing Instructions:**

1: Yes.

---

### Decision · Program_Chairs · 2022-05-15

**Decision:**

Accept (Poster)

**Comment:**

Meta Review: The paper proposes a mechanism for learning sparse networks by combining multiple sub-networks.
Overall the reviewers appreciated the contribution and empirical results.
In the discussion, the issue of comparison to SWA came up. As noted there, the authors are asked to more clearly highlight the difference between their method and an SWA approach that simply learns sparse models and averages those.
Is the key difference that their method can allow some of the averaged models to be non-sprase?
A more detailed explanation and ablation would be very helpful for demonstrating the advantage of the method.